# Plant phenomics & precision agriculture simulation of winter wheat growth by the assimilation of unmanned aerial vehicle imagery into the WOFOST model

Tianle Yang[1,2,3☯¤], Weijun Zhang[1,2,3☯¤], Tong Zhou[1,2,3¤‡], Wei Wu[1,2,3¤‡], Tao Liu[1,2,3¤], Chengming Sun[1,2,3¤] *

1 Jiangsu Key Laboratory of Crop Genetics and Physiology/Jiangsu Key Laboratory of Crop Cultivation and Physiology, Agricultural College of Yangzhou University, Yangzhou, China, 2 Jiangsu Co-Innovation Center for Modern Production Technology of Grain Crops, Yangzhou, China, 3 College of Agriculture, Yangzhou University, Yangzhou City, Jiangsu Province, People's Republic of China

☯ These authors contributed equally to this work.
¤ Current address: Department of Agricultural Information Technology, Agricultural College of Yangzhou University, Yangzhou City, Jiangsu Province, People's Republic of China
‡ These authors also contributed equally to this work.
* cmsun@yzu.edu.cn

**Data Availability Statement:** All relevant data are within the paper and its Supporting Information files.

## Abstract

The aim of this study is to optimize the simulation result of the WOFOST model and explore the possibility of assimilating unmanned aerial vehicle (UAV) imagery into this model. Field images of wheat during its key growth stages are acquired with a UAV, and the corresponding leaf area index (LAI), biomass, and final yield are experimentally measured. LAI data is retrieved from the UAV imagery and assimilated into a localized WOFOST model using least squares optimization. Sensitive parameters, i.e., specific leaf area (SLATB0, SLATB0.5, SLATB2) and maximum $CO_2$ assimilation rate (AMAXTB1, AMAXTB1.3) are adjusted to minimize the discrepancy between the LAI obtained from the model simulation and inversion of the UAV data. The results show that the assimilated model provides a better estimation of the growth and development of winter wheat in the study area. The $R^2$, RMSE, and NRMSE of winter wheat LAI simulated with the assimilated WOFOST model are 0.8812, 0.49, and 23.5% respectively. The $R^2$, RMSE, and NRMSE of the simulated yield are 0.9489, 327.06 kg·hm$^{-2}$, and 6.5%. The accuracy in model simulation of winter wheat growth is improved, which demonstrates the feasibility of integrating UAV data into crop models.

## Introduction

The coupling of crop models and remote sensing data is a novel approach in modern agricultural research and involves multiple fields, such as agriculture, mathematics, and remote sensing. Crop modeling is an effective tool in the simulation of crop growth. It reflects the causal

**Funding:** National Natural Science Foundation of China (31701355,31671615), National Postdoctoral Foundation (2016M600448,2018T110560), Classification of Project (2016YFD0300107), and Dominant Disciplines of Jiangsu Province (2017).

**Competing interests:** The authors have declared that no competing interests exist.

relationship between genetic pattern, regulated cultivation, and environmental conditions in crop growth. However, as only discrete spatial data at the regional scale is obtained in the models, the spatial correlation is weak, which limits the simulation accuracy of models over large regions [1]. Remote sensing is capable of making synchronous observations over large areas and possesses strong real-time characteristics, which is lacking in the crop models [2]. Crop simulation models with remote sensing data assimilation can dynamically reflect the inherent growth mechanism of crops while incorporating global real-time dynamic monitoring capabilities of remote sensing [3]. Showing good prospect in regional crop yield prediction, this approach has attracted the attention of agricultural scientists both in China and overseas [4] and is one of the central topics and research directions in the study of information-based agriculture, anticipated to remedy the problems and deficiencies of using crop modeling or remote sensing alone [5].

Data assimilation was initially developed for numerical weather prediction [6], mostly in ocean environments [7]. Its application has gradually expanded to land surface data [8,9]. Since Wiegand [10] et al. first proposed the introduction of the vegetation index into crop models to improve the accuracy of model simulation, scientists have looked extensively into the feasibility of integrating remote sensing information into crop models [11,12], which has laid the foundation for the assimilation of remote sensing data into crop models. Data assimilation is achieved primarily by the forcing method, updating method, or parameter optimization method. Numerous studies have shown that the parameter optimization method is superior to the forcing and updating methods in terms of the assimilation results. This method involves the coupling of different data and the constant updating of the initial values of the model parameters according to the selected mathematical algorithms until the optimal set of model parameters is found. It is used to minimize the difference between the simulated data and actual measurements [13]. Compared to the other two methods, the parameter optimization method is less affected by the time, space, and errors of remote sensing observations. It allows for errors in remote sensing data and is the most widely used assimilation method.

Research on the integration of crop models with remote sensing has mainly focused, on the acquisition of remote sensing data, optimization algorithms and the parameters to be optimized. Presently, with the ongoing effort into assimilation research, the acquisition of remote sensing information has developed from using visible and near infrared light in the early days to thermal infrared and microwave. Assimilation methods are also evolving [14]. However, most studies deal with the assimilation of high-altitude platform-based remote sensing and spectral remote sensing into crop models, while few investigate the assimilation of unmanned aerial vehicle (UAV) platform-based remote sensing into crop models. UAV technology is increasingly used in agricultural monitoring because of the lower manpower input required and fewer interference factors. In this paper, a localized WOFOST model and leaf area index (LAI) retrieved from UAV data are assimilated using a parameter optimization method based on the least squares algorithm. The simulation results with assimilation are validated in the hopes of laying the foundation for the joint use of UAV and modeling.

## Materials and methods

### Field experiment design

**Experiment 1.** This part of the study, i.e. experiment 1, was performed in the experimental field of Yangzhou University during 2015–2016. The crop variety of winter wheat tested was Yangmai 19. The crop grown previously was rice. The soil type was sandy loam containing 24.65 g·kg$^{-1}$ organic matter, 106.34 mg·kg$^{-1}$ hydrolyzable nitrogen, 122.09 mg·kg$^{-1}$ available phosphorus, and 89.74 mg·kg$^{-1}$ available potassium in the 0–20 cm soil layer.

Three crop density levels were set: 150·m$^{-2}$ (M1), 225·m$^{-2}$ (M2), and300·m$^{-2}$ (M3). Four nitrogen levels were set: 0 kg·ha$^{-1}$ (N1), 180 kg·ha$^{-1}$ (N2), 240 kg·ha$^{-1}$ (N3), and 300 kg·ha$^{-1}$ (N4). In terms of fertilizer application, nitrogen fertilizer was applied as a base fertilizer, fertilizer for the tillering stage, fertilizer for the jointing stage, and fertilizer for the booting stage in the proportion of 5:1:2:2, while phosphorus and potassium fertilizers were applied as a base fertilizer and during the jointing stage in the proportion of 5:5. The amount applied was 120 kg·ha$^{-1}$ for all fertilizers. Seeds were sown on November 2, 2015 over an 18m$^2$ plot and a replicate plot of 10.8 m$^2$. Field experiments were conducted in 2016 on January 7, March 7, March 27, April 12, and May 30 to measure and calculate the wheat yield and LAI.

**Experiment 2.** This part of the study was performed in 2016–2017. The crop varieties tested were Yangmai 23 and Ningmai 13. The proceeding crop was rice. The soil type was sandy loam containing 23.14 g·kg$^{-1}$ organic matter, 106.12 mg·kg$^{-1}$ hydrolyzable nitrogen, 101.08 mg·kg$^{-1}$ available phosphorus, and 87.78 mg·kg$^{-1}$ available potassium per 0–20 cm of soil layer.

Three crop density levels were set: 150·m$^{-2}$ (M1), 225·m$^{-2}$ (M2), and 300·m$^{-2}$ (M3). Four nitrogen levels were set: 0 kg·ha$^{-1}$ (N1), 180 kg·ha$^{-1}$ (N2), 240 kg·ha$^{-1}$ (N3), and 300 kg·ha$^{-1}$ (N4). In terms of fertilizer application, nitrogen fertilizer was applied as base fertilizer, fertilizer for tillering stage, fertilizer for jointing stage, and fertilizer for booting stage in the proportion of 5:1:2:2, while phosphorus and potassium fertilizers were applied as base fertilizer and during jointing stage in the proportion of 5:5. The amount applied was 120 kg·ha$^{-1}$ for all fertilizers. Seeds were sowed on November 14, 2016 over a 9.8 m2 plot. Field experiments were conducted in 2017 on January 6, February 28, March 14, April 24, and May 19 to measure and calculate the wheat yield and LAI.

## Evaluation of WOFOST parameters

The model was validated in two ways: by the comparison of scatter plots and the quantitative evaluation of simulated and measured values with selected statistical evaluation indices. Scatter plots for the regression analysis of simulated and measured LAI, biomass, and yield were constructed. The coefficient of determination ($R^2$), coefficient of residual mass (CRM), root mean square error (RMSE), and normalized root mean square error (NRMSE) were chosen as the statistical evaluation indices. $R^2$ quantifies the consistency between the measured and simulated data; a value close to 1 reflects good simulation results. CRM could be positive or negative; a positive value indicates that the model overestimates the measurements, whereas a negative value indicates underestimation. RMSE and NRMSE show the relative and absolute errors, respectively, between simulated and measured data; smaller RMSE and NRMSE values indicate better simulation results. NRMSE $\leq$ 10% is obtained only on extremely high-accuracy simulations. 10% < NRMSE $\leq$ 20% indicates high accuracy, 20% < NRMSE $\leq$ 30% indicates medium accuracy, and NRMSE > 30% means the accuracy is low [15,16]. These indices were calculated using Eqs 1–4:

$$R^2 = \frac{SSR}{SST} = 1 - \frac{SSE}{SST} \tag{1}$$

$$CRM = \frac{\sum_{i=1}^{n} Y_i - \sum_{i=1}^{n} X_i}{\sum_{i=1}^{n} X_i} \tag{2}$$

$$RMSE = \sqrt{\frac{1}{n}\sum_{i=1}^{n}(Y_i - X_i)^2} \tag{3}$$

$$NRMSE = \frac{RMSE}{\bar{X}} \times 100\% \tag{4}$$

where SSR is the regression sum of squares, SST is the total sum of squares, SSE is the error sum of squares, $i$ is the $i$th sample, $Y_i$ and $X_i$ are the simulated and measured values of the $i$th sample, respectively, $\bar{X}$ is the average of the measured values of all samples, and $N$ is the number of samples.

## Assimilation of UAV imagery for modeling

Using assimilation, the optimal values are determined for the sensitive parameters to be optimized. These values minimize the discrepancy between model-simulated LAI and LAI retrieved from UAV image data, as Eq 5 shows:

$$\int(I_1, I_2, I_3, I_4, I_5) = \sum_{i=1}^{n}(LAI_M - LAI_{UAV}) \tag{5}$$

$I_1$, $I_2$, $I_3$, $I_4$, and $I_5$ in the equation denote the parameters to be optimized, i.e. SLATB0, SLATB0.5, SLATB2, AMAXTB1, and AMAXTB1.3. $LAI_M$ and $LAI_{UAV}$ refer respectively to the LAI simulated by WOFOST model and from the inversion of UAV imagery. n is the number of LAI data assimilated each time, which is 5.

## Least squares method

In recent years, a number of algorithms have been applied to the assimilation of remote sensing data for modeling in China and worldwide, such as using the Kalman filter [17,18] and simulated annealing algorithm [19]. While demonstrating good adaptability in different studies, a more general application of these algorithms is difficult due to the complexity of the computation involved. The least squares method is a mathematical optimization technique. It finds the function that best matches the data by minimizing the sum of the squares of the differences between the observed values and estimated values. This method is moderately complex and is widely used in many fields. Using the least squares method, the unknown data could be easily obtained, and the sum of squares of the differences between the obtained data and the actual data are minimized. Therefore, it is adopted here as the optimization algorithm to construct the cost function. By calculating the differences between model simulation results and UAV image inversion results, a most appropriate set of parameters is determined for optimization, which minimizes the discrepancy between simulation and observation. The cost function is shown in Eq (6).

$$F(x) = \sum_{i=1}^{n}(LAI_M - LAI_{UAV})^2 \tag{6}$$

where $LAI_M$ and $LAI_{UAV}$ refer respectively to the LAI simulated by the WOFOST model and from the inversion of UAV imagery. $n$ is the number of LAI data assimilated each time, which is 5.

To ensure reasonable parameter optimization and avoid optimized parameters that are mathematically sound but do not make practical sense, the optimization process is ended when one of the following three conditions is met: 1) the optimization function is not improved by at least 0.0001% after five consecutive cycles; 2) the number of iterations exceeds

10000; 3) the parameters for optimization all reach their threshold values. The algorithm is shown in S1 Fig.

## Data assimilation process

In this study, UAV image data were assimilated for modeling, following the procedure shown in S2 Fig.

A model for the retrieval of winter wheat LAI from UAV imagery was constructed. The best estimated LAI value given by the model is marked as $LAI_{UAV}$. Data sample sets established by enforcing uniform distribution of errors were constructed for the five parameters, SLATB0, SLATB0.5, SLATB2, AMAXTB1 and AMAXTB1.3, which were identified by sensitivity analysis for optimization. Other parameters were adjusted as required by the model (e.g., TSU-MEM,TSUM1,TSUM2,CVS). The WOFOST model was recursively called to simulate the LAI values for each of the five growth stages. For all experimental settings, the optimum parameter values were found using the cost function as the ones giving the least errors, and the optimum range of values for each of the five parameters was determined (Table 1).

## Simulation results of the model with UAV data assimilation

A comparison is made in S3 Fig between (1) Model-simulated LAI with and without assimilation and (2) LAI retrieved from UAV data. UAV imagery was used in this study to retrieve LAI for data assimilation into models. The localized model refers to the adjustment of related parameters so that the model can simulate the growth of wheat in the study area. Comparing the model-simulated winter wheat LAI with and without assimilation, it was observed that the localized model and model with assimilation both adequately reflect the growth and development of winter wheat. LAI reached its maximum value at around 148 days after seeding. Differences are clearly seen in the graph between model-simulated LAI with and without assimilation. At low density and low nitrogen level, LAI values simulated with and without assimilation show significant differences. At high density and high nitrogen levels, the differences are smaller. Nevertheless, for all experimental settings, the parameter values assimilated by least squares optimization are able to correct for the model and give LAI estimations that are closer to the actual values.

## Validation of the WOFOST model with assimilation

### Validation of simulated LAI

Winter wheat LAI simulated by the WOFOST model with assimilation was validated by measured LAI in the study area for 2016–2017, as shown in S4A Fig scatter plot of LAI simulated by the WOFOST model with assimilation and measured LAI are graphed in this case. As shown in S4 Fig, the calibrated WOFOST model gives satisfactory simulation results on winter wheat LAI. $R^2$ is 0.8812, close to the 1:1 line. This indicates good agreement between

**Table 1. Initial values and optimized values of sensitive parameters.**

| Parameter for optimization | Initial value | Optimized value |
|---|---|---|
| SLATB0 | 0.001–0.00212 | 0.00125–0.002 |
| SLATB0.5 | 0.00125–0.00215 | 0.0011–0.0018 |
| SLATB2 | 0.0007–0.0012 | 0.0007–0.001 |
| AMAXTB1 | 38.83 | 35–45 |
| AMAXTB1.3 | 30–60 | 38–60 |

simulation and measurements. This $R^2$ is also higher than the value for LAI simulated by the localized WOFOST model (0.8178), which demonstrates a better agreement between simulation and measurements. The RMSE of model-simulated LAI with assimilation is 0.49, and the NRMSE is 23.5% (20% < NRMSE ≤ 30% indicates medium accuracy). These errors fall within the confidence interval. The RMSE and NRMSE also show a decrease of 0.09 and 4.4% compared with their respective values (0.58 and 27.9%) for LAI modeled with localization only. This means that the simulation errors of the model are reduced by assimilation and the simulation accuracy is improved. The CRM of model-simulated LAI with assimilation is −0.12, and CRM < 0 indicates overall smaller simulated values than measurements. In conclusion, the WOFOST model with assimilation gives more accurate estimation on the LAI of winter wheat, and the simulation results are generally better than the localized WOFOST model.

## Validation of simulated yield

**Yield comparison for different experimental settings.** Winter wheat yields simulated by the WOFOST model with assimilation are compared with measured yield data in the study area under different experimental settings for 2016–2017. As shown in S5 Fig, the simulation results of the model with assimilation essentially follows the same trend as the actual yield across all experimental settings. Except for the yields of M2N1 and M3N1, which are close to the measured values, and that of M1N2, which is slightly above the actual value, the yields simulated by the WOFOST model with assimilation are in general lower than the actual ones, but are above the estimations given by the localized model at high nitrogen levels. Moreover, the error between simulated yield and actual yield decreases after assimilation. For high nitrogen levels in particular, the original model shows large error in yield simulation. This is suppressed significantly by assimilation.

Winter wheat yields simulated by the WOFOST model with assimilation were validated by measured yield data in the study area during 2016–2017. Scatter plots of the simulated and measured values were created, as shown in S6 Fig. In this figure, the calibrated WOFOST model gives overall satisfactory estimation of winter wheat yield. $R^2$ is 0.9489, close to the 1:1 line, indicating good consistency between simulation and measurements. This value is also clearly higher than the $R^2$ of yield simulated by the localized WOFOST model (0.5852), demonstrating a substantial improvement in the agreement between the simulation results and measurements. With assimilation, the RMSE of the model-simulated yield is 327.06 kg·hm$^{-2}$ and the NRMSE is 6.5%. These errors are within the confidence interval and are reduced by 472.9 kg·hm$^{-2}$ and 9.4% from the respective RMSE and NRMSE of yields estimated by the localized model (799.96 kg·hm$^{-2}$ and 15.9%, respectively). The errors in the model-simulated yield with assimilation were significantly lower, and the NRMSE was below 10%, i.e., the simulation accuracy improved from moderate to high accuracy. The CRM of the model-simulated yield with assimilation is −0.04, where CRM < 0 indicates that the simulated values are generally higher than the measured values. In conclusion, the WOFOST model makes better prediction of winter wheat yield after assimilation.

## Discussion

The assimilation of remote sensing data for crop modeling improves the accuracy of model simulation and allows for the accurate and timely input of parameter values acquired in regional-scale models [20,21]. In this sense, the employment of crop growth models at the regional scale could be realized via the assimilation of spatially continuous remote sensing data into temporally continuous crop models to monitor the temporal and spatial continuity of crop growth [22]. Many studies have been performed on the assimilation of satellite remote

sensing data into models [1,5,14,17,19], but few exist on the assimilation of UAV remote sensing data for modeling [23]. The use of unmanned aerial vehicles in agriculture is also relatively recent. Compared to satellite remote sensing, UAV remote sensing is still under development. However, to expand the application of crop models from a single location to an area and a region, it is necessary to integrate UAV remote sensing into modeling. Data assimilation is demonstrated by this study to be a viable strategy for the realization of this integration.

Data assimilation can be achieved in many ways. Three data assimilation methods are commonly used for modeling purposes:, the forcing method, updating method, and parameter optimization method, of which the parameter optimization method is the most effective. The least squares method is a generic optimization algorithm that is frequently used in many fields. It was employed by Zhao et al [24]. to retrieve winter wheat LAI with time-series HJ CCD remote sensing data as the external data source and study data assimilation into models. Their results showed that the error in simulated LAI was reduced by 4.98% compared to the results obtained from unassimilated data, and the error in simulated yield was lowered by 2.28%. This demonstrates the applicability of least squares method as an assimilation algorithm. The results we obtained on the assimilation of UAV imagery into the WOFOST model are also consistent with those of previous studies [25], in that the simulation accuracy of the crop model is improved by data assimilation.

This study aims to optimize the WOFOST simulation results and the improvement of yield simulation accuracy for winter wheat by the assimilation of LAI retrieved from UAV imagery into the model. However, it was discovered during parameter optimization that both directly input parameters and LAI retrieved from UAV are prone to the influence of many factors, such as planting density, amount of fertilizer applied, and population size. In future research, crop growth should be comprehensively evaluated from multiple aspects, and model parameters should be optimized using multi-factor constraints, to improve the accuracy and generality of the model.

## Supporting information

**S1 Fig. Least squares optimization process.**
(TIF)

**S2 Fig. Data assimilation process.**
(TIF)

**S3 Fig. Comparison of LAI simulated with localization, with assimilation and LAI retrieved from UAV data.**
(TIF)

**S4 Fig. Comparison between LAI simulated with assimilation and measured LAI.**
(TIF)

**S5 Fig. Comparison of simulated yields obtained under different experimental settings.**
(TIF)

**S6 Fig. Comparison of yields simulated with assimilation and measured yields.**
(TIF)

**S1 Dataset.**
(XLS)

## Author Contributions

**Conceptualization:** Tao Liu.

**Data curation:** Tianle Yang.

**Formal analysis:** Weijun Zhang.

**Project administration:** Tao Liu, Chengming Sun.

**Resources:** Tong Zhou.

**Software:** Wei Wu.

**Writing – original draft:** Tianle Yang.

**Writing – review & editing:** Tianle Yang.

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
