## [Decision Letter · Decision Letter 0]

13 Nov 2020

PONE-D-20-28606

Simulation of winter wheat growth by the assimilation of unmanned aerial vehicle imagery into the WOFOST model

PLOS ONE

Dear Dr. Sun,

Thank you for submitting your manuscript to PLOS ONE. After careful consideration, we feel that it has merit but does not fully meet PLOS ONE’s publication criteria as it currently stands. Therefore, we invite you to submit a revised version of the manuscript that addresses the points raised during the review process.

There is need for more information for making the WOFOST modelling more transparent to the reader by adding the used parameters and settings.In addition, the authors should describe in detail the extraction of LAI from UAV imaging and how accurate that process it.

We look forward to receiving your revised manuscript.

Kind regards,

Abel Chemura

Academic Editor

PLOS ONE

Journal Requirements:

2. Our internal editors have looked over your manuscript and determined that it is within the scope of our Call for Papers on Plant Phenomics & Precision Agriculture. This collection of papers is headed by a team of Guest Editors for PLOS ONE. Additional information can be found on our announcement page: https://plos.io/phenomics.

If you would like your manuscript to be considered for this collection, please let us know in your cover letter and we will ensure that your paper is treated as if you were responding to this call. If you would prefer to remove your manuscript from collection consideration, please specify this in the cover letter.

3. Please ensure that you include a title page within your main document. We do appreciate that you have a title page document uploaded as a separate file, however, as per our author guidelines (http://journals.plos.org/plosone/s/submission-guidelines#loc-title-page) we do require this to be part of the manuscript file itself and not uploaded separately.

Reviewers' comments:

Reviewer's Responses to Questions

**Comments to the Author**

1. Is the manuscript technically sound, and do the data support the conclusions?

Reviewer #1: Yes

Reviewer #2: Partly

2. Has the statistical analysis been performed appropriately and rigorously? 

Reviewer #1: Yes

Reviewer #2: I Don't Know

3. Have the authors made all data underlying the findings in their manuscript fully available?

Reviewer #1: Yes

Reviewer #2: No

4. Is the manuscript presented in an intelligible fashion and written in standard English?

Reviewer #1: Yes

Reviewer #2: Yes

5. Review Comments to the Author

Reviewer #1: his paper represents a study of simulation of winter wheat growth by the assimilation of unmanned aerial vehicle imagery into the WOFOST model at the finer spatial resolution. And readers are very interested in the topic. Generally, this paper is well written, some comments can be found as followed,

(1) Suggest conduct an sensitivity analysis to determine the optimized parameters, I think that specific leaf area (SLATB0, SLATB0.5, SLATB2) and maximum CO2 assimilation rate (AMAXTB1, AMAXTB1.3) is not suitable for reinitialized because CO2 assimilation rates are relative constant values for wheat.

(2) I didn’t see how the retrieve the LAI from UAV imagery and evaluation of accuracy of LAI. More texts need to be added to clarify this point.

(3) The uncertainty of observation and model in the assimilation method. This actually to be very important for improving the performance of DA. More texts need to be added to clarify this issue.

(4) I suggest that don’t cite the Chinese papers in the manuscript, because English readers are not easy to understand the content of Chinese papers. Meanwhile added the latest DA literature to support your viewpoint. (1) Huang et al.,. Assimilation of remote sensing into crop growth models: current status and perspectives. Agricultural and Forest Meteorology. 276-277, 2019,107609. (2) Jianxi Huang et al., Evaluation of regional estimates of winter wheat yield by assimilating three remotely sensed reflectance datasets into the coupled WOFOST–PROSAIL model. European Journal of Agronomy. 2019,102, pp.1-13;(3) Jianxi Huang et al., Assimilating a synthetic Kalman filter leaf area index series into the WOFOST model to estimate regional winter wheat yield. Agricultural and Forest Meteorology. 216, 2016. pp.188-202. (4) Jianxi Huang, Hongyuan Ma, Wei Su, Xiaodong Zhang， Yanbo Huang, Jinlong Fan, Wenbin Wu. Jointly assimilating MODIS LAI and ET products into the SWAP model to estimate winter wheat yield. IEEE Journal of Selected Topics in Applied Earth Observations and Remote Sensing. 8(8), 2015.pp. 4060--4071. (5) Wen Zhuo et al., Assimilating Soil Moisture Retrieved from Sentinel-1 and Sentinel-2 Data into WOFOST Model to Improve Winter Wheat Yield Estimation. Remote Sens. 2019, 11(13), 1618

(5) English of this paper need to be further polished by professional editor.

Reviewer #2: Review for Plos One

Title : Simulation of winter wheat growth by the assimilation of unmanned aerial vehicle imagery into the WOFOST model

1. Introduction

It is written in lines 48-50 that the parameter optimization method is superior in terms of assimilation results. This statement should be clearly explained and supported by a bundle of citations. To the knowledge of the Reviewer, other methods such as updating method using EnKF have quite high performance especially in the consideration of the measurement uncertainties. In addition, the operability of the proposed assimilation method should be adequately discussed? When do we acquire data to assimilate them during the growing season?

The originality of this contribution relies on the fact that, although UAV is widely used in crop monitoring, most studies related to assimilation methods use high-altitude remote sensing (RS) and not UAV data. The interest of UAV-based RS should be better highlighted in terms of scales, operability, spatial and temporal resolution with respect to, for instance, satellite-based RS.

In the introductive section, the proposed research work should be better detailed and supported by the state of the art in order to specify the scientific objectives with regard to the considered material and methods. For the Reviewer, the formulation of the paper objectives are too general.

2. Material and methods

The material and methods section appears not complete enough to better understand the research scope and limitations of the proposed method. Why two experiments are considered in response to the research objectives? The measurement methods for LAI and crop yield are not described at all, which renders difficult the interpretation of the results. Also, please give the crop density in terms of grains per meter square.

Although WOFOST is quite known in the community, the key principle of modelling, the considered assumptions and values of major parameters (line 191) should be given for the sake of completeness and research reproducibility. Also, please give the definition of R2 in Equation 1.

The extraction of LAI from UAV imaging is not described. This is a critical point to explain, present and discuss in terms of methodology, operability and accuracy. What are the sensors used? What is the altitude? What is the regression? …

It is not clear for the Reviewer when data assimilation is performed. Is it at each remotely sensed LAI? If so, this should be better highlighted via a Figure. Is it necessary to perform assimilation at all measurement point?

Equation 4 and 5 are not mathematically valid. Please adapt them.

Line 199, please correct “A comparison in made in …”

Line 203, please define earlier in the paper the notion of localized model.

Line 227, please give the definition of the confidence interval.

Line 273, CRM is negative

3. Discussion

The scale of the study should be criticized. The Authors present their model as valid at the regional scale. Is it realistic with UAV technology?

6. PLOS authors have the option to publish the peer review history of their article (what does this mean?). If published, this will include your full peer review and any attached files.

Reviewer #1: No

Reviewer #2: No

---

## [Author Response · Author response to Decision Letter 0]

6 Jan 2021

Reviewer #1: his paper represents a study of simulation of winter wheat growth by the assimilation of unmanned aerial vehicle imagery into the WOFOST model at the finer spatial resolution. And readers are very interested in the topic. Generally, this paper is well written, some comments can be found as followed,

(1)Suggest conduct an sensitivity analysis to determine the optimized parameters, I think that specific leaf area (SLATB0, SLATB0.5, SLATB2) and maximum CO2 assimilation rate (AMAXTB1, AMAXTB1.3) is not suitable for reinitialized because CO2 assimilation rates are relative constant values for wheat.

R: The parameters for initialization adjustment in this article are all determined by the results of sensitivity analysis. Taking into account the local varieties and cultivation measures, compare the leaf area and the maximum CO2 assimilation rate (AMAXTB1, AMAXTB1.3) for reinitialization to make the results more accurate.

(2) I didn’t see how the retrieve the LAI from UAV imagery and evaluation of accuracy of LAI. More texts need to be added to clarify this point.

R: The extraction of LAI from UAV images and the content of LAI accuracy evaluation have been explained in detail in another article.

(3) References

R: Replaced

Reviewer #2: Review for Plos One 

1. Introduction 

It is written in lines 48-50 that the parameter optimization method is superior in terms of assimilation results. This statement should be clearly explained and supported by a bundle of citations. To the knowledge of the Reviewer, other methods such as updating method using EnKF have quite high performance especially in the consideration of the measurement uncertainties. In addition, the operability of the proposed assimilation method should be adequately discussed? When do we acquire data to assimilate them during the growing season? The originality of this contribution relies on the fact that, although UAV is widely used in crop monitoring, most studies related to assimilation methods use high-altitude remote sensing (RS) and not UAV data. The interest of UAV-based RS should be better highlighted in terms of scales, operability, spatial and temporal resolution with respect to, for instance, satellite-based RS. In the introductive section, the proposed research work should be better detailed and supported by the state of the art in order to specify the scientific objectives with regard to the considered material and methods. For the Reviewer, the formulation of the paper objectives is too general.

R: (1) The parameter optimization method mentioned in this article is actually a general term for EnKF, SAA, and ordinary least squares. In the research, ordinary least squares were selected. (2) This study mainly carried out data acquisition in the five key stages of wheat growth: jointing stage, heading stage, flowering stage, filling stage, and mature stage.

2. Material and methods 

The material and methods section appears not complete enough to better understand the research scope and limitations of the proposed method. Why two experiments are considered in response to the research objectives? The measurement methods for LAI and crop yield are not described at all, which renders difficult the interpretation of the results. Also, please give the crop density in terms of grains per meter square. Although WOFOST is quite known in the community, the key principle of modelling, the considered assumptions and values of major parameters (line 191) should be given for the sake of completeness and research reproducibility. Also, please give the definition of R2 in Equation 1. The extraction of LAI from UAV imaging is not described. This is a critical point to explain, present and discuss in terms of methodology, operability and accuracy. What are the sensors used? What is the altitude? What is the regression? … It is not clear for the Reviewer when data assimilation is performed. Is it at each remotely sensed LAI? If so, this should be better highlighted via a Figure. Is it necessary to perform assimilation at all measurement point? Equation 4 and 5 are not mathematically valid. Please adapt them. Line 199, please correct “A comparison in made in …” Line 203, please define earlier in the paper the notion of localized model. Line 227, please give the definition of the confidence interval. Line 273, CRM is negative

R: (1) The data of the second year is used to verify the results; (2) The leaf area index measurement method: 15 plants were sampled in the experimental plot, and the length and width coefficient method was used to determine the leaf area, and then the specific leaf weight method was used to calculate the leaf area index. Yield measurement: In the mature period of wheat, the number of ears per unit area was investigated in each plot, and 30 single ears were taken for indoor experiment to calculate the three factors of yield (number of ears, number of grains per ear, and thousand-grain weight). Take 1m2 area of wheat from each plot, thresh and test the yield. (3) In this study, the inspire 1RAW drone produced by DJI is equipped with a Zenmuse X5R aerial camera for data collection. The flying height of the drone is 9 meters. (4) What is regression (5) The localized model refers to the adjustment of related parameters so that the model can simulate the growth of wheat in the study area. (6) This study mainly carried out data acquisition and assimilation in five key stages of wheat growth: jointing stage, heading stage, flowering stage, filling stage, and mature stage.

3. Discussion 

The scale of the study should be criticized. The Authors present their model as valid at the regional scale. Is it realistic with UAV technology?

R: The combination of UAV remote sensing and WOFOST model is applied to the simulation of crop agronomic parameters. This method is still in the exploratory stage. Cheng et al. used a combination of multispectral UAV and WOFOST model to estimate corn yield. This proves that this method is feasible. (Cheng Z, Meng J, Shang J, et al. Generating Time-Series LAI Estimates of Maize Using Combined Methods Based on Multispectral UAV Observations and WOFOST Model[J]. Sensors, 2020, 20(6006):6006.)

---

## [Decision Letter · Decision Letter 1]

28 Jan 2021

Simulation of winter wheat growth by the assimilation of unmanned aerial vehicle imagery into the WOFOST model

PONE-D-20-28606R1

Dear Dr. Sun,

We’re pleased to inform you that your manuscript has been judged scientifically suitable for publication and will be formally accepted for publication once it meets all outstanding technical requirements.

Kind regards,

Abel Chemura

Academic Editor

PLOS ONE

Additional Editor Comments (optional):

Reviewers' comments:

Reviewer's Responses to Questions

**Comments to the Author**

1. If the authors have adequately addressed your comments raised in a previous round of review and you feel that this manuscript is now acceptable for publication, you may indicate that here to bypass the “Comments to the Author” section, enter your conflict of interest statement in the “Confidential to Editor” section, and submit your "Accept" recommendation.

Reviewer #1: All comments have been addressed

2. Is the manuscript technically sound, and do the data support the conclusions?

Reviewer #1: Yes

3. Has the statistical analysis been performed appropriately and rigorously? 

Reviewer #1: Yes

4. Have the authors made all data underlying the findings in their manuscript fully available?

Reviewer #1: Yes

5. Is the manuscript presented in an intelligible fashion and written in standard English?

Reviewer #1: Yes

6. Review Comments to the Author

Reviewer #1: The authors made a good revision following reviewers's comments and the manuscript is greatly improved.

The manuscript can be accepted for publication in present form.

7. PLOS authors have the option to publish the peer review history of their article (what does this mean?). If published, this will include your full peer review and any attached files.

Reviewer #1: No

---

## [Editor Report · Acceptance letter]

30 Sep 2021

PONE-D-20-28606R1 

Plant Phenomics & Precision Agriculture
Simulation of winter wheat growth by the assimilation of unmanned aerial vehicle imagery into the WOFOST model 

Dear Dr. Sun:

I'm pleased to inform you that your manuscript has been deemed suitable for publication in PLOS ONE. Congratulations! Your manuscript is now with our production department. 

Kind regards, 

on behalf of

Dr. Abel Chemura 

Academic Editor

PLOS ONE